# Effect of Dietary Supplementation on Milk Nutrient Deposition and Enteric Methane Emission in Dual-Purpose Cows from the Colombian Amazon

**DOI:** 10.3390/ani15243542

**Published:** 2025-12-09

**Authors:** Brandon Stiven Bustamante Castaneda, Linda Estefanía Rodríguez Hernández, Paula Andrea Méndez Santos, Anderson Ferlay Aldana Novoa, Maira Alejandra Mejía Sánchez, Ligia Johana Jaimes Cruz, Diana María Bolivar Vergara, Juan Evangelista Carulla Fornaguera, Rolando Barahona Rosales, Jésus Alfredo Berdugo-Gutiérrez, Sorany Milena Barrientos Grajales, John Jairo Montoya Zuluaga, Iván Darío Pino Giraldo, Juan Manuel Cerón Alzate, María Victoria Galeano, Darío Antonio Timarán Vallejo, Héctor Jairo Correa Cardona

**Affiliations:** 1Department of Animal Production, Universidad Nacional de Colombia, Carlos E Restrepo, Medellín 050034, Colombia; bsbustamantec@unal.edu.co (B.S.B.C.);; 2Department of Animal Production, Universidad Nacional de Colombia, Bogotá 111311, Colombia; lrodriguezh@unal.edu.co (L.E.R.H.);; 3Facultity of Medicine Veterinary and Zootech, Institución Universitaria Visión de las Américas, Medellín 050030, Colombia; ligia.jaimes@uam.edu.co; 4Universidad Nacional de Colombia, Vía Caño Limón, San Nicolas, Arauca 810001, Colombia; 5Department of Animal Science, Universidad Nacional de Colombia, Vía Candelaria, Palmira 763531, Colombia; 6Technical Department, Productora de Insumos Agropecuarios SOMEX SAS, Guayabal, Medellín 050023, Colombia; 7Technical Department, Cooperativa COLANTA, Medellín 050034, Colombia; 8Escuela Superior de Medicina Veterinaria y Zootecnia, Cholula de Rivadavia 72770, Puebla, Mexico; 9Centro de Investigación Obonuco, Corporación Colombiana de Investigación Agropecuaria, km 5 vía Pasto to-Obonuco, Pasto 520038, Colombia; davallejo@agrosavia.co

**Keywords:** ARDUINO, concentrate, mineralized salt, proteinized salt, spirometry-mask

## Abstract

Methane emitted by cattle is a major contributor to greenhouse gas emissions from livestock systems. However, accurately quantifying these emissions under grazing conditions remains a challenge, especially in tropical regions where specialized infrastructure is limited. The present study evaluated an innovative spirometry mask prototype designed to measure enteric methane emissions directly from grazing cattle in field conditions. This mask integrates a low-cost MQ4 methane sensor and data recording system that allows continuous measurements during normal animal activity. Through experimental trials, we demonstrated that the spirometry mask can reliably estimate methane emissions while maintaining animal comfort and normal behavior. This development provides a practical and accessible alternative for researchers and producers to monitor emissions in extensive production systems, contributing to more sustainable livestock management and the development of mitigation strategies adapted to tropical environments.

## 1. Introduction

Cattle farming is an important economic activity in tropical countries because of its contribution to food security through meat and milk production. In Colombia, the per capita milk intake is 154 L/person/year, while beef intake is 18.2 kg/person/year [1]. These amounts represent a substantial proportion of the milk and meat intake recommended by the WHO [2]. Colombia currently has a cattle inventory of approximately 29 million head, distributed across a variety of production systems that include breeding, fattening, dual-purpose (DPS) farming, and specialized dairy farming [3]. These systems are present throughout the national territory and take advantage of the diverse geographical and climatic conditions of each region. DPS, in particular, has expanded notably in recent years and now provides the majority of the country’s annual milk production [4], making it highly relevant within the national herd due to its contribution to job and income generation [5]. These production systems are characterized by the use of crossbreeding between *Bos indicus* and *Bos taurus* breeds to improve adaptation to tropical conditions, enabling producers to obtain economic benefits from the sale of milk and weaned calves in local, national, and international markets [6,7]. They are also based on grazing tropical grasses with low levels of feed supplementation, and cows are partially milked to allow calves access to residual milk [8].

Grazing on tropical grasses forms the basis of feeding under the DPS, but it faces challenges due to seasonality in the availability and quality of the pastures which, in turn, generates low milk production on a seasonal basis [9].

Tropical grasses use the C4 phosphoenolpyruvate carboxykinase metabolic path-way for carbon fixation [10,11], which leads to a greater accumulation of structural carbo-hydrates and lignin compared to those that use the C3 pathway [12,13]. It can affect the palatability and digestibility of these forages in ruminants [13,14] and limits the ability of animals to extract essential nutrients efficiently, and frequently cause protein, energy, and mineral deficiencies [15]. As a consequence, the intake of forage by animals is affected and the production of enteric methane (CH_4_) per unit of dry matter intake is increased, thus reducing productive efficiency [12,16,17]. Grazing on grasses that use the C4 metabolic pathway, compared to grazing on those that use the C3 pathway, is associated with a lower efficiency in milk and meat production in ruminants, partially due to an increase in CH_4_ emissions [17,18,19].

The generation of gases in the rumen is closely related to the microbial activity that occurs during fermentation. CH_4_ produced in this process represents about 6.5% of the gross energy consumed [20], although values of 11.4% [21] and up to 13.7% [22] have been reported. This figure can vary significantly depending on the animal’s diet, with lower levels found in diets rich in grains and higher levels found when low-quality forage is consumed [23,24].

Besides the importance of CH_4_ in the energy efficiency of animals, its role as a greenhouse gas within the scientific landscape over the last 50 years should be considered. The global warming potential of this gas was calculated to be 28 times that of carbon dioxide [25]. Consequently, there has been a sustained increase in research on CH4 production in the gastrointestinal tracts of ruminants, as well as in the development of strategies to reduce these emissions [26].

Dietary supplementation stands out among the strategies suggested to reduce enteric CH4 emissions in ruminants [27]. However, this response depends on the types and amounts of supplements used [26,28,29]. Therefore, it is necessary to evaluate different types of dietary supplements in the field and their efficiency in the milk deposition of nutrients, as well as their ability to improve milk production and reduce CH_4_ emissions. The most commonly used food supplements in the Colombian tropics include mineralized salts, multi-nutritional blocks, and concentrated foods [30,31]. However, there are no reports on the effects of these supplements on CH4 emissions in lactating cows under dual-purpose systems in the Colombian low tropics. This paucity of research may be due to the lack of methods for measuring CH4 emissions from grazing livestock that are easy to implement, economical, and provide detailed information on this process. However, Correa and Jaimes [32] recently proposed a prototype portable electronic spirometry mask that allows one to measure the emissions of CH_4_ eliminated by exhalations. Therefore, the objective of this experiment was to quantify the effect of three dietary supplements on the milk production, milk deposition of nutrients, and enteric CH_4_ emissions in lactating cows grazing in a dual-purpose system located in the Colombian low tropics.

## 2. Materials and Methods

### 2.1. Ethics

The Ethics Committee in Animal Research of Universidad Nacional de Colombia at Medellín (Antioquia) approved this study as part of the research projects under HERMES 57438, “Interfaculty alliance for the reduction in greenhouse gases in cattle farming” (CICUA-18-23).

### 2.2. Location

This study was carried out from 19 April to 6 May 2024, in a commercial property under a dual-purpose production system in the municipality of San Vicente del Caguán (2°06′55″ N, 74°46′12″ W), located in the Colombian Amazon region, department of Caquetá. This area is located at an altitude of approximately 280 masl and classified as a humid to very humid tropical forest [33]. The region has an average annual temperature of 25.2 °C, with a relative humidity that exceeds 70% and an average annual rainfall of 2518 mm. This area is characterized by abundant rainfall and good quality soil. Moreover, dominant pastures in the area are composed of a variety of pastures, with Mombasa (*Panicum maximum cv. Mombasa*) being the most abundant, followed by the Humidicola (*Brachiaria humidicola*) and Star grass (*Cynodon nlemfuensis*) pastures.

### 2.3. Animals and Experimental Design

Twenty-four F1 Holstein × Gyr multiparous lactating cows *(Bos taurus* × *Bos indicus*) were selected. These cows had an average live weight of 485.95 ± 55.6 kg during the second third of lactation, with an approximate milk production of 10.48 ± 3.22 L of milk per day. The cows had an average age of 81 ± 20.02 months with 192.2 ± 92.3 days in lactation and an average body condition of 3.12 ± 0.45, according to a rating scale from 1 to 5, where 5 indicates obesity and 1, emaciation [34]. The cows continued to graze primarily on Mombasa grass and star grass under a rotational system, with approximately 32 days of rest and two to three days of grazing with strips of ap-proximately 2200 m2/day being offered each morning after milking. The size of grazing strips offered was limited with an electrical fence.

The experiment was structured as a randomized complete block design (Figure 1) with four consecutive 12-day experimental periods (blocks), in which cows were as-signed to one of three treatments (food supplements). Food supplements were prepared at the San Pablo Agricultural Station of Universidad Nacional de Colombia at Medellín, located in Rionegro, Antioquia. These supplements were formulated based on the average chemical composition of predominant pastures in the experimental field. Likewise, the assigned doses of each supplement were determined considering the estimated nutritional needs of cows [34] and the analyses results of the pastures. Experimental treatments consisted of mineralized salt (MS) offered at a rate of 100 g/cow/day; proteinized salt (PS) offered at a rate of 400 g/cow/day; and concentrated feed (CO) offered at a rate of 1 kg per 3 L of milk/cow/day. Each supplement was given in two equal servings/day during each of the two milkings (approximately 6:00 a.m. and 5:00 p.m.). Table 1 presents the composition of raw materials and chemical analysis of these supplements.

### 2.4. Sample Size

The sample size of 24 cows (8 cows per treatment) was determined based on herd availability and the operational capacity of the methane-measurement system, with four temporal blocks and two cows per treatment per block. Because enteric methane was the primary response variable of interest, a post hoc power evaluation was calculated using the observed variability of daily methane production. The mean CH_4_ emission was 195.67 g/day, with a within-treatment standard deviation of 17.03 g/day. Assuming α = 0.05, two-sided tests, three treatments, and 8 cows per treatment, the minimal detectable difference with 80% power was ap-proximately 1.4 × SD (23.8 g CH_4_/day). The study had sufficient statistical power to detect (12% of the mean emission level) moderate changes in daily methane pro-duction under grazing conditions.

### 2.5. Data Collection

The 24 cows randomly entered the experiment in four blocks of 12-day experimental periods with a two-day interval between blocks and six cows/block in a randomized complete block design. Therefore, the block is one of four groups of six cows that entered the experiment every two days. In each block, two cows were randomly assigned to one of the three experimental treatments (CO, PS and MS). On the first day of each experimental period, the food supplement received by the animals was withdrawn, and the milk production of each cow was determined. Based on the results, cows were assigned in a balanced manner to each of the three treatments in such a way that in each period, there were two cows/treatment. Subsequently, the cows began an adaptation period to the treatments, lasting until day 10. Between day 4 and 7, each cow was fitted with a polypropylene halter as part of the adaptation period to the measurement equipment (see Figure 2A). Between day 8 and 10, each cow was fitted with a halter to which a prop mask that simulated the measurement equipment was assembled. However, this mask featured only a plastic box installed on one side, a mask that covered the nostrils, and a hose covered by foam to avoid damage to the equipment due to possible blows (see Figure 2B). On days 11 and 12, between 7:00 a.m. and 4:00 p.m., the cows were fitted with a halter including a portable electronic spirometry mask (ESM) that completely covered the animals’ nostrils (see Figure 2C). This mask featured a CH_4_ sensor (MQ4, Hanwei Electronics, Zhengzhou, China), an airflow sensor (DN-40, Louchen ZM Shenzhen,, China), and an exhalation counter similar to that described by CorreaCorrea and Jaimes [32] and were calibrated as described by Jaimes et al. [36]. These three components were connected to an ARDUINO board (model ONE R3 MEGA328P, Si Tai&SH Hengtai Store, Hong Kong, China) programmed using ARDUINO IDE 2.0.4 to store the information on a MicroSD card every 500 ms.

Likewise, pasture sampling was conducted during the last three days of each experi-mental period (day 10 to day 18 in Figure 1), before the cows entered the pasture strip. A proportional amount of each type of grass present in the strip was taken at a height of approximately 10 cm. These samples were mixed to obtain a final sample, which was then stored under refrigeration for chemical analysis. Elsewhere, during the last three days of each experimental period samples were taken from the experimental supplements, and from feces during each milking. These samples were mixed to obtain a single sample/animal and kept under refrigeration until they were brought to the laboratory for analysis.

During each of the two milking, cows were handled in individual cubicles with a feeder, and a supplement was fed in the amount corresponding to each treatment. When necessary, the hind legs of the cows were immobilized to avoid damage to the measuring equipment or accidental injury to the animals or operators. Milking was stimulated by the presence of calves and lasted between 7 and 10 min; during this time, supplements were offered.

### 2.6. Milk Production and Milk Sampling

The milk production of each cow was measured from both milking during the last three days of each period and the mean of this measurements was used to the statistical analysis. Approximately 50 mL of milk samples was taken from each animal every milking. Then, these samples were mixed to obtain a single representative sample per animal. This mixed sample was frozen for subsequent analysis in the laboratory. On days of milk production measurement, calves were present alongside the cows only to stimulate milking. Subsequently, the calves were separated to accurately record the amount of milk produced under each supplement.

### 2.7. Estimated Dry Matter Intake (DMC)

To estimate the dry matter intake (DMI) of forage, indigestible dry matter (iDM) was used as an internal marker and chromium oxide (Cr_2_O_3_) as an external marker [37]. The iDM of the forage, supplements, and feces of each experimental cow was determined using an in vitro test for 144 h with feces taken directly from the recta of three Holstein cows adapted to a diet based on Kikuyu grass (*Pennisetum clandestinum*) and concentrated feed [38] as an inoculum. To estimate the production of feces, chromium oxide (Cr_2_O_3_) was used as an external marker [39]. This marker was prepared in a 1:1 ratio with the concentrated feed and then pelletized to ensure intake and minimize losses. Each animal was given 15 g of these pellets during each milking for the entire experimental period. In the feces samples collected twice per day on the last three days of each experimental period, the concentration of Cr, as well as in the pelletized mixture, was determined via atomic absorption spectrometry. A marker recovery rate of 80% was assumed according to the results obtained by Rueda and Correa [40] in a dual-purpose farm in the department of Cesar. Feces production (F) was estimated following the formula proposed by Lippke [41]:F, _g_ = (g of Cr in feed) × (Cr recovery rate in feces) /% fecal Cr

The dry matter intake from pasture (PDMI) was estimated with the formula proposed by Geerken et al. [42] using information on indigestible dry matter in feces (iDMf), food supplements (iDMs), and pastures (iDMp), as well as information on F and dry matter intake from supplements (DMIs):PDMI_(kg/cow/d)_ = ((iDMf × F) − (iDMs × DMIs))/iDMf

Total DMI (TDMI) was estimated for each cow as the sum of PDMI and supplement dry matter intake (SDMI), which was determined daily as the difference between the quantity offered and the quantity rejected.

### 2.8. Laboratory Analysis

The contents of fat, crude protein, total solids, and lactose in milk samples were determined via infrared spectroscopy using the standard method ISO 9622, IDF 141:2013 [43]. Calcium (Ca) and potassium (K) were determined with atomic absorption spectrometry, while phosphorus (P) was measured using UV–VIS spectrophotometry. Milk production was adjusted considering the fat content, standardizing it to 4.0% fat (FCM) [44]. In feces, the content of Ca, K, and Cr was determined with atomic absorption spectrometry and P with UV–VIS spectrophotometry. Nitrogen (N) was quantified using the Kjeldahl volumetric method, and the neutral detergent fiber (NDF) was measured with the gravimetric method [45]. The presence of C, P, and K was quantified in the mineral salt, while the quantity of ash (Ash), Ca, P, N, and K was determined in proteinized salt. Finally, in the mixture of grasses from pasture and the concentrated supplement, gross energy was determined using an adiabatic bomb calorimeter (IKA^®^ C5000, Cole Palmer, Vernon Hills, IL, USA) and NDFs were evaluated with the methods described by Van Soest and Robertson [46], as was ethereal extract, N, Ash, Ca, P, and K [45].

### 2.9. Apparent Digestibility

The apparent digestibility of NDF, N, P, Ca, and K was determined from the amount of each fraction consumed and the amount determined in feces. The efficiency of nutrients in milk deposition was calculated for N, P, K, and Ca as the percentage of each nutrient consumed that was secreted in the milk produced (g/cow/day).

### 2.10. Calculation of CH4 Emissions

The amount of CH_4_ (CH_4_) emitted by each animal through the exhaled air was determined by the following formula [32]:CH_4_, L/min = Ve × CH_4_c/1,000,000
where Ve is the volume of air exhaled (L/min), and CH_4_c is the concentration of CH_4_ (ppm).

To express enteric CH_4_ emissions in grams, a conversion factor of 1 L of CH_4_ = 0.716 g was applied [47]. Based on this information, the yield (L of CH_4_/kg of DMI), intensity of CH_4_ emissions (L of CH_4_/L of milk), and fraction of consumed EB transformed into CH_4_ (Y_m_(%)) were calculated. The intensity of CH_4_ emissions was also calculated in proportion to the level of milk production corrected for fat (MF) and protein (MP) (FPMP) [48]:FPCM (kg) = raw milk (kg) × (0.337 + 0.116 × MF (%) + 0.06 × MP (%))

### 2.11. Statistical Analysis

Response variables were analyzed using the PROC MIXED procedure in SAS v 8.0 (SAS Inc. 1999, Release 8.0 SAS Inst., Inc., Cary, NC, USA) by applying the following mixed model:Y_ijk_ = µ + B_i_ + C_k_(B)_j_ + S_k_ + ε_ijk_
where Y_ijk_ is the dependent variable, µ is the mean of all observations, B_i_ is the fixed effect of block i, C_k_(B)_i_ is the random effect of cow k within block i, S_k_ is the fixed effect of supplement l (PS, MS, and CO), and ε_ijk_ is the random residual error. Mean analysis was carried out with LSMEANS test using the SAS statistical package. Differences were considered significant at *p* ≤ 0.05.

## 3. Results

One dairy cow in the MS treatment was removed from the experiment because she stopped producing milk before the trial was completed. The live weight of the animals was not affected by the treatments. The results for dry matter intake, nutrient intake, and apparent digestibility are reported in Table 2. Here, PDMI and TDMI were not affected by the experimental treatments (*p* > 0.05) but, GEI was higher in CO (*p* < 0.05). Since PDMI did not differ between treatments and remained the main source of NDF and K, its intake also did not differ between treatments. In turn, Ca intake was higher with CO and PS (*p* > 0.001), while P intake was higher with PS (*p* < 0.002). Finally, N intake was higher with CO (*p* < 0.01). The above results indicate that due to differences in the nutritional composition and supplements evaluated, as well as the amounts supplied, the supplements exerted different effects on the intake of chemical fractions and nutrients evaluated.

Table 3 presents the results for the production, nutritional composition, and efficiency in the deposition of nutrients in the experimental cows’ milk production. Ultimately, there was no effect of the treatments on milk production and nutritional composition. However, the deposition efficiency of Ca and P in milk was significantly higher under the MS and CO treatments (*p* < 0.04), with no effect of the treatments on the deposition efficiency of N and K in milk.

To further analyze and discuss our results, a Pearson correlation analysis was carried out between milk production and efficiency in the deposition of nutrients (Table 4). As can be seen, correlations between milk production and efficiency in the deposition of Ca, P, K, and N in this experiment were positive, suggesting that an increase in milk yield significantly improved nutrient efficiency (*p* < 0.01)

Data on the CH_4_ emissions are summarized in Table 5, which indicate that treatments did not affect any of the expressions in which emissions were quantified in this experiment, except to percentage of GEI losses as CH_4_ (Y_m_) that was lower in CO (*p* < 0.05). However, their values agree with those reported in the literature.

## 4. Discussion

The chemical composition of the pastures used in this experiment is within the values reported for tropical grasses, where high levels of NDF and low CP are notable [49,50]. Likewise, the GE content of the pastures is within the values reported for tropical grasses, with values ranging between 3.8 [50] and 4.13 Mcal/kg of DM [51]. Meanwhile, the GE content of the concentrated supplement found in this work is similar to that reported by Robles [52] in an experiment carried out in Mexico with dual-purpose grazing cows.

The response of grazing animals to dietary supplementation depends on the amount and availability of nutrients in the supplement, as well as their ability to correct the limitations of the pasture, whose effects can be reflected in the digestibility, intake, and metabolism of nutrients [53]. Carulla et al. [54] also noted that the amount of food an animal consumes is a determining factor in its productive performance. The authors stated that approximately 70% of differences in the productivity of grazing animals are attributable to variations in their food intake, with the availability and quality of forage playing a fundamental role. In this experiment, the evaluated food supplements did not affect the PDMI or the TDMI, although the ADMD was higher with the CO treatment (Table 2). This treatment (CO) had a high starch content (approximately 40%) and true protein (approximately 16.9%) but its intake was relatively low (approximately 0.9 kg DMI/Cow/d) and, by this, it could have an improve the microbial growth and, then, digestible fraction [55,56]. Thus, Palma et al. [55] and Lazzarini et al. [56] reported an increase in digestibility of organic matter when animals are supplemented with starch and protein supplement but no observed an effect on dry matter intake. Although higher ADMD is generally expected to increase PDMI [57] and exert a substitution effect [58], this relationship is highly variable, which may explain the lack of TDMI response observed here. Previous studies report that even marked changes in ADMD do not necessarily translate into changes in TDMI [59,60]. Bargo et al. [58] also highlight the wide variability in dietary responses to concentrate supplements in fibrous diets, which depend on supplement characteristics, forage quality, and animal factors. Similar to our findings, several authors report no effect of nitrogenous salt or concentrate supplementation on TDMI in dual-purpose cows grazing tropical forages [61]. In some cases, ADMD also remained unaffected [61]. Evidence from high-producing cows further supports that salt supplementation does not consistently increase TDMI [62]. Overall, the TDMI values obtained in this study fall within the ranges previously reported for comparable production systems [40,63,64]. GEI (Mcal/cow/d) was higher in CO treatment (Table 2) due to higher gross energy content of this supplement (Table 1) and due to this intake was higher (approximately 0.9 kg/cow/d) than PS (approximately 0.4 kg/cow/d) and MS (approximately 0.1 kg/cow/d).

The digestibility of NDF (NDFD) was not affected by treatments (Table 2) despite differences in the nutritional composition of supplements and amounts consumed. Although supplementation with starchy foods such as that used in this experiment could reduce the NDFD, this reduction occurs only when supplementation levels are high [58]. In this situation, the fermentability of starches increased the volatile fatty acids production reducing pH [65,66], the fibrolytic bacterial population [66] and colonization of fiber [67], but increasing passage rate from rumen resulting in a reduction in retention time and fiber fermentation in rumen [68]. All this explain the substitution rate between supplement and forage when the supplementation is medium to high [69]. Bargo et al. [69] found no effect of supplementation with 4.0 kg/cow/d of a concentrated feed on the NDFD when the supply of forage was limited. Gutierrez et al. [61] also observed no changes in the NDFD in cows supplemented with nitrogenous salt ad libitum or a concentrated supplement (up to approximately 3.0 kg/cow/d). Other authors have likewise reported that supplementation with mineralized salt [70] or multi-nutritional blocks does not affect the DNDF [71]. The NDFD values found in this experiment (Table 2) are among the values reported by Ismartoyo et al. [72] in diets based on *Panicum maximum* but lower than the value reported by Relling et al. [73] when evaluating the same grass during three seasons of the year, with three ages of regrowth.

Although the quantity of Ca and P in the MS was high (Table 1), the amount of the supplement given to and consumed by the animals was not sufficient, and negative apparent digestibility was observed for these minerals (Table 2). Negative apparent digestibility of Ca and P has been reported in cattle consuming low-quality tropical forages and reflects a mismatch between low dietary mineral intake and substantial endogenous mineral fluxes. In the case of Ca, its availability is limited by strong binding to structural components of the plant cell wall, such as uronic and phenolic acids, and by the formation of insoluble Ca–oxalate complexes common in tropical grasses [74]. These interactions reduce ruminal and post-ruminal release of Ca and also trap endogenously secreted Ca, increasing its fecal output beyond intake [75,76,77,78]. For P, apparent digestibility is even more sensitive to dietary concentration because ruminants recycle large quantities of P through saliva, and 95–98% of endogenous P is excreted in feces [34]. Consequently, when forage P is low, fecal P is dominated by endogenous sources, producing negative digestibility values, as confirmed by radiolabeled studies showing up to ~70% endogenous contribution [79]. Therefore, negative apparent digestibility of Ca and P does not indicate net loss of body reserves but rather reflects low dietary supply, extensive endogenous secretion, and mineral binding to indigestible forage fractions.

The lack of treatment effects on PDMI and TDMI (Table 2) was reflected in milk yield (Table 3), consistent with findings from Gutierrez et al. [61] and Robles et al. [52] in dual-purpose and crossbred cows grazing tropical pastures. Because milk production is closely linked to DMI, as widely documented in both tropical [80] and temperate systems [81], the absence of changes in intake likely explains the unchanged milk yield. Moreover, milk production often varies independently of supplementation due to genetic potential, lactation stage, forage availability, and management factors [82]. Although Ca and P intake increased with the PS treatment (Table 2), this did not affect PDMI, TDMI, or milk production (Table 3). Similar lack of response has been reported in studies evaluating dietary P inclusion or mineral supplementation in dairy cows [83,84].

Likewise, no effects of supplementation were observed on milk composition, in agreement with reports from Gutierrez et al. [61], Robles et al. [52], and Aguilar-Pérez et al. [85], among others [86,87]. Even in high-producing cows, increasing concentrate levels does not generally modify milk composition, as shown by Lawrence et al. [88] and Muñoz et al. [89].

For salt supplementation, the results of this study agree with those of Li et al. [90], who also found no effects on the production and composition of milk from Holstein cows in China after supplementation with mineralized salt. Pal et al. [91] also found no differences in the content of fat, lactose, protein, and non-fat solids in the milk of half-breed cows in India supplemented with salt compared to a control group. Wu et al. [92] likewise found no effect of P concentration on the composition of milk in Holstein cows fed with a fully mixed ration. Similar results have been reported with supplementation with multi-nutritional blocks. Valk and Kogut [84] observed no effect of supplementation with five mineralized blocks on the composition of milk, while Verma et al. [93] also reported no effect of supplementation with proteinized salt on the production and composition of milk in buffaloes in India. These results may be due to the fact that the responses to supplementation in the production and composition of milk are variable and depend on multiple factors such as the genetics of the animal, the stage of lactation, the availability and quality of the pastures, and management practices [84,94,95], all of which can be reflected in the digestibility, intake, and metabolic use of nutrients [53] and, therefore, in the production and composition of milk.

In modern dairy farms, achieving greater efficiency in the use of nutrients has become essential to improve profitability, considering that feed represents about 50% of total costs and is key for cows to express their genetic potential in milk production [96]. This efficiency also has a direct impact on the environmental impact of livestock systems, which is especially relevant in the face of growing concerns about climate change [97]. Considering the challenge of feeding a constantly growing world population, it has become necessary to optimize the production per unit of nutrients instead of expanding the use of resources, thereby prioritizing the sustainability of the system [98]. In this project, the efficiency in the use of four nutrients was quantified, three of which have been associated not only with the environmental impacts generated by livestock systems in various parts of the world but also because these nutrients are considered essential for the development of sustainable pasture-based livestock systems, such as N, P, and K [99]. The results obtained show that CO and MS supplementation improved the efficiency of Ca (*p* < 0.04) and P (*p* < 0.009) (Table 3) milk deposition. The average values were slightly higher than those reported by Rueda and Correa [40] in a dual-purpose herd of cows from the Department of Cesar, whose values ranged between 10% and 13.9% for Ca and between 17.0% and 19.9% for P. These values, in turn, are lower than those reported by other authors for cows in different latitudes. Thus, Taylor et al. [100] reported that deposition efficiency of Ca in the milk of Holstein cows is higher than 31.6%, while Aarons et al. [101] reported that said efficiency in Australian grazing cows can range between 8 and 76%. For P, this efficiency ranges between 4 and 48%, without explaining the factors that determine said variation. In this work, the correlations between milk production and efficiency in the deposition of Ca (r = 0.836) and P (r = 0.881 in milk (Table 4) were positive and significant (*p* < 0.001), thereby suggesting that improving milk production has positive effects on the use efficiency of nutrients such as Ca and P, as in the present study.

The efficiency of K deposition in milk in this study was not affected by the treatments, with values ranging between 5.81 and 6.48% (see Table 3). These low efficiencies, however, have been reported by other authors in lactating cows under different production systems. Jaimes et al. [102] found that the efficiency in the use of this mineral in Holstein cows in northern Antioquia was 6.4%, while Rueda and Correa [40] reported efficiencies between 6.4 and 9.6% in a dual-purpose tropical farm in Colombia. Likewise, the efficiency in the deposition of N in milk in this work was not affected by treatments, with averages similar to those reported by Jaimes et al. [102] in northern Antioquia with Holstein cows. The authors observed an average of 20.7%, while Rueda and Correa [40] reported slightly higher values between 26.9 and 32.2%. These results could be due to differences in nutrient content in the diet, food intake, and milk production [99].

Table 4 shows positive correlations between the efficiency of Ca, P, K, and N in milk deposition with milk production. This result suggests that by increasing milk yield in animals, the use of these nutrients for milk production can be improved. This effect represents the basis for improving the efficiency in nutrient use within milk production systems worldwide for more than a century [103], based on the principle that increasing milk production proportionally reduces the animal maintenance requirements [104]. Likewise, Table 4 shows the negative correlations between the intake of each nutrient and the milk deposition efficiency of those same nutrients. This phenomenon occurs because reducing the supply of a mineral increases its efficiency, as mentioned by Arriaga et al. [104], which argued that the best strategy to increase mineral use efficiency is to decrease its proportion in the diet.

The importance of CH_4_ as a greenhouse gas has led to researching and developing strategies that reduce its emissions in the world’s livestock systems [105]. CH_4_ emissions have been expressed in various ways with different explanations and implications for production systems management. In ruminants, such emissions can be evaluated using three indicators: CH_4_ production, expressed as the amount emitted per animal and per day (g/d); CH_4_ intensity, measured per unit of product obtained (milk or meat, g/kg); and CH_4_ yield, calculated per kilogram of dry matter intake (DMI) and expressed in g/kg [106]. This performance can also be related to the intake of organic matter, fiber in neutral detergent [107], or digestible organic matter [108]. According to de Haas et al. [106], CH_4_ production depends mainly on food intake, linked to body size and milk productivity. Intensity, on the other hand, is conditioned by the volume of milk and energy requirements, while performance reflects the methanogenic potential of the diet. The relevance of each indicator varies depending on the production system and economic context.

In this study, the means of enteric CH_4_ concentrations ranged between 1732 and 1911 ppm (see Table 5), without being affected by treatments. These values are within the ranges reported by Washburn and Brody [109] and Koch et al. [110], who used spirometry masks that completely covered the snouts of lactating cows. In the present study, CH_4_ production was not affected by treatments and averaged 196 ± 18 g/cow/d, a value consistent with previous reports for lactating cows in tropical systems [111]. The lack of differences in methane production among the CO, PS, and SM treatments can be explained by the similarity in total nutrient supply, fermentation patterns, and forage utilization across treatments. The pastures offered in this study had a chemical composition typical of tropical grasses, characterized by high NDF and low CP, and their gross energy values were within the expected range for C4 forages (Table 1). Under these conditions, the response to supplementation depends largely on whether supplements correct major nutritional limitations of the forage and stimulate changes in digestibility, intake, or rumen metabolism [53]. In this experiment, despite the differences in nutrient composition among supplements, neither PDMI nor TDMI differed among treatments, indicating that supplements did not modify overall feed intake (Table 2). Others authors [112,113] report low to zero correlation between nutritional composition of diet to cattle with CH_4_ production. Because CH_4_ output in grazing cattle is strongly driven by total dry matter intake [114,115], the absence of change in PDMI and TDMI largely explains the lack of differences in CH_4_ emissions.

The Y_m_ values found in this experiment are similar to those reported by Pozo et al. [116] but slightly lower than those found by Rivera et al. [64] in dual-purpose production systems under tropical conditions. In this experiment, Y_m_ was lower with CO (6.30%) and higher with PS (8.23%), primarily because the GEI was higher with CO but lower with PS (*p* < 0.05) (Table 2), while there were no differences in CH_4_ production between treatments (*p* > 0.15) (Table 4). These results reflect differences in the energy efficiency of ruminal fermentation associated with the type of supplement. Now, although with CO there was less loss of energy consumed, this was not reflected in milk production (Table 3), which is difficult to explain due to the complexity of energy transactions in the rumen and in the animal, which do not allow a clear relationship to be established between the energy lost as CH_4_ and the productive response of the animal [117].

Although the CO treatment increased apparent digestibility of dry matter (Table 3), the amount of supplement consumed was low (~0.9 kg DMI/cow/d), and therefore insufficient to shift the ruminal fermentation profile in a way that would materially influence CH_4_ production. Previous studies have shown that small amounts of starch or protein supplements can improve digestibility without altering total intake [55,56], and that changes in digestibility do not always translate into changes in methane [118] if total substrate flow to the rumen remains unchanged. Likewise, NDF digestibility was not affected by supplementation. Thus, reductions in fiber digestibility—and the associated decreases in methane—are expected only when starch supplementation is high enough to depress ruminal pH and inhibit fibrolytic bacteria, which was not the case here. The low supplementation levels also explain the absence of substitution effects on forage intake, maintaining similar total fermentable organic matter across treatments.

Taken together, these results indicate that the supplementation strategies evaluated did not meaningfully alter the quantity or fermentability of the substrate entering the rumen. Since methane production in ruminants is closely associated with fermentable organic matter intake and ruminal fermentation patterns, the similarity in PDMI, TDMI, NDFD, and overall nutrient supply across treatments resulted in comparable methane emissions. This aligns with previous reports showing that, in tropical grazing systems, supplementation at modest levels often improves nutrient balance or specific digestibility components without necessarily modifying the ruminal fermentation intensity or total organic matter intake required to influence methane output.

In the present study, CH_4_ production was not affected by treatments and averaged 196 ± 18 g/cow/d, a value consistent with previous reports for lactating cows in tropical systems [111]. The lower emissions compared with those reported by Rivera et al. [64] may be due to differences in forage digestibility, as the higher dry matter digestibility (DMD) observed by those authors (55–60%) would increase fermentable substrate availability in the rumen, thereby enhancing methanogenesis. In contrast, the lower DMD in the current work (46 ± 8.2%) would have constrained ruminal fermentation and reduced CH_4_ yield per unit of intake. Methodological factors may also contribute, since polytunnel systems quantify total CH_4_ emissions, whereas the equipment used here measures only cranial emissions, potentially underestimating total output. Comparisons with Primavesi et al. [119] and Villanueva et al. [120] similarly highlight that higher CH_4_ production in other studies is associated with greater intake of highly digestible supplements, which increases ruminal fermentability and microbial activity, ultimately elevating CH_4_ production alongside milk yield. This is consistent with the observed parallel increases in CH_4_ output and milk production across studies, reflecting the shared dependence of both processes on the supply of digestible nutrients and the intensity of ruminal fermentation.

Overall, these results indicate that variation in CH_4_ emissions among tropical dairy systems is primarily driven by differences in diet digestibility, supplement quality and quantity, and methodological approach, and that improvements in nutrient supply can enhance milk production while concurrently altering methanogenic potential.

The average intensity of CH_4_ emissions found in this experiment was 35.5 ± 12.4 g/L of milk produced, without any effect of the treatments on this variable (*p* < 0.35). This average is within the values reported by other authors. Primavesi et al. [119] documented an intensity of 25.3 g/L of milk for grazing mongrel cows that consumed 7.6 kg of guinea grass MS (*P. maximun*), were supplemented with 3.4 kg/cow/d of concentrated feed, and produced 13.3 L of milk/cow/d. On the other hand, Yassegoungbe et al. [121] observed values between 97.4 and 111.5 g of CH_4_/L of milk in cows belonging to small local breeds of Benin (Africa) such as white Fulani, Boboji, and Yakana, whose milk production ranged from 1.1 to 1.2 kg/cow/d. Villanueva et al. [120], however, recorded values of 16.09 g/L of milk. The above data show an inverse relationship between milk production and CH_4_ emission intensity, as indicated by Corredu et al. [122] and Boshe et al. [123], who reported a negative correlation of −0.25 and −0.65, respectively, between these two variables.

The intensity of CH_4_ emission expressed by FPCM has a linear relationship with milk production, so the previous discussion applies to this expression of CH_4_ emissions. Finally, CH_4_ emissions (g/kg TDMI), like the other variables, were not affected by the experimental treatments due to the absence of effects on dry matter intake (see Table 2) and CH_4_ production (see Table 5).

## 5. Conclusions

The dietary supplements evaluated (CO, PS, MS) in this experiment did not enhance milk yield, milk composition, or reduce enteric CH_4_ emissions in grazing dual-purpose cows. Therefore, this could be because the effect of supplementation on the dry matter intake on the other variables is multifactorial and affected by elements such as the quality and quantity of supplements, quality and availability of forage, and characteristics of the animal. However, CO and MS improved the efficiency of Ca and P deposition in milk, offering potential benefits for nutrient-use efficiency in tropical grazing systems. Finally, supplementation with concentrate showed less losses of gross energy as methane (Y_m_) due possibly, to best fermentative efficiency.

## Figures and Tables

**Figure 1 animals-15-03542-f001:**
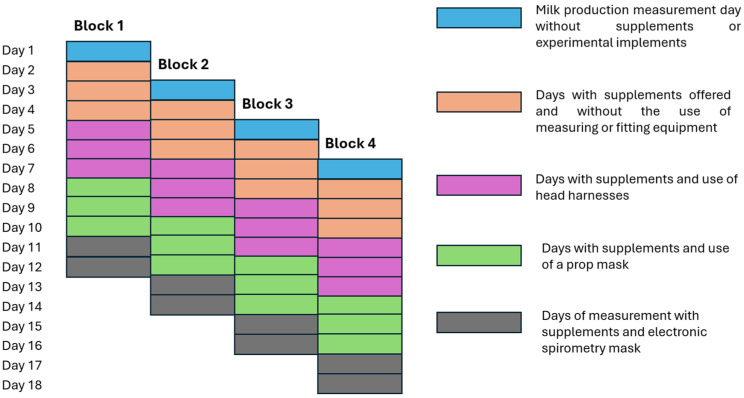
Schematic showing experimental periods, duration of each experimental period, and total duration of the experiment. Source: authors.

**Figure 2 animals-15-03542-f002:**
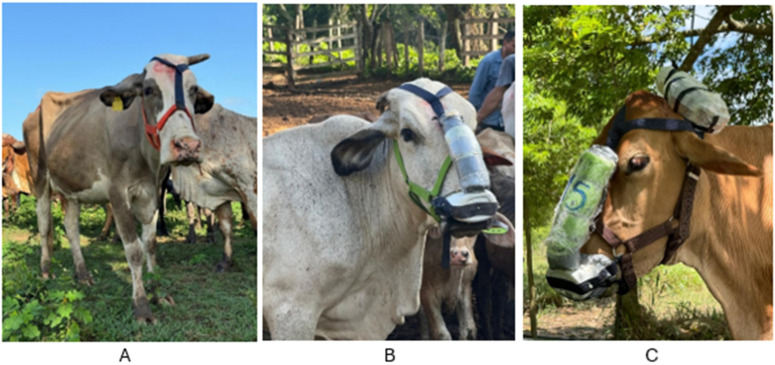
Adaptive and measuring harnesses (spirometry mask): (**A**) halter, (**B**) halter with prop mask, and (**C**) halter with electronic spirometry mask. Source: authors.

**Table 1 animals-15-03542-t001:** Ingredients and chemical composition of experimental pastures and food supplements.

	^1^ MP	MS	PS	CO
Ingredients, kg/Ton				
Corn				500
Soybean cake			330	240
Wheat bread				100
Palm kernel				100
Molasses		50	50	49
Bran		80.5	100	
Ca carbonate		192	113	10
Dicalcium phosphate		360	210	
NaCl		240	70	
Sulphur power		25	22	
Mg oxide		40	40	
Urea			60	
Premix		12.5	5	1
Chemical composition, % ^2^ DM				
NDF	69.0			15.2
N	1.50		5.40	2.70
Ash	9.26		37.6	5.88
Ca	0.27	13.67	7.87	1.08
P	0.18	7.45	4.78	0.39
K	1.50	2.24	1.16	1.02
iDM	53.9	ND	16.8	13.1
GE, Mcal/kg	4.04		3.14	4.42
ME, ^3^ Mcal/kg	1.39			^4^ 3.12
NEl, ^5^ Mcal/kg	0.791			^5^ 2.00

^1^ MP: grass mix in grasslands; MS: mineralized salt; PS: proteinized salt; CO: concentrated feed. ^2^ DM: dry matter; NDF: neutral detergent fiber; N: nitrogen; Ash: total ash; Ca: calcium; P: phosphorus; K Potassium; iDM: indigestible dry matter at 144 h; GE: gross energy. ^3^ ME (metabolizable energy) = 0.17 × DMi − 2.0 [35]. ^4^ ME (metabolizable energy) = 1.01 × ED − 0.45. ^5^ NE_l_ (net energy) = 0.703 × EM − 0.19 [34].

**Table 2 animals-15-03542-t002:** Intake of dry matter, calcium, potassium, phosphorus, and nitrogen in dual-purpose cows from San Vicente del Caguán (Caquetá) supplemented with mineralized salt (MS), proteinized salt (PS), and concentrated feed (CO).

Response Variables	Treatments			SE	*p*
	^1^ CO	PS	MS		
^2^ LW, kg	494	529	470	25.0	0.281
PDMI, kg/day	6.83	7.53	8.83	0.536	0.067
TDMI, kg/day	9.53	7.89	8.93	0.536	0.116
GEI, Mcal/day	39.6 ^a^	31.7 ^b^	35.7 ^ab^	2.16	0.050
NDF, kg/day	5.20	5.19	6.09	0.370	0.213
N, g/d	183 ^a^	134 ^b^	132 ^b^	0.008	0.001
Ca, g/d	50.8 ^a^	51.9 ^a^	37.6 ^b^	0.001	0.001
K, g/d	153	140	161	0.009	0.375
P, g/d	24.0 ^b^	32.7 ^a^	23.1 ^b^	0.001	0.001
Apparent Digestibility %					
DM	53.2 ^a^	41.1 ^b^	41.9 ^b^	2.25	0.002
NDF	45.7	43.2	47.2	2.42	0.521
N	59.5 ^a^	51.3 ^ab^	38.8 ^b^	3.11	0.001
Ca	26.4 ^a^	10.4 ^a^	−25.6 ^b^	8.88	0.001
K	74.2	78.5	73.9	2.29	0.483
P	11.2 ^a^	18.7 ^a^	−15.0 ^b^	4.41	0.001

Means in the same row, followed by different superscripts (^a^, and ^b^), are significantly different (*p* < 0.05); ^1^ CO: concentrate; PS: proteinized salt: MS: mineralized salt; *p*: Statistical significance value; SE: Standard error; ^2^ LW, kg: live weight in kg; PDMI: forage dry matter intake; TDMI: total dry matter intake; GEI: gross energy intake; NDF: neutral detergent fiber; N: nitrogen; Ca; calcium; K: potassium; P: phosphorus; DM: dry matter. Table 2 also shows apparent digestibility. Here, the apparent digestibility of DM (ADMD) was higher under the CO treatment (*p* < 0.003), while NDF digestibility was not affected by the treatments. Likewise, the apparent digestibility of N, Ca, and P was similar between the CO and PS treatments but higher than that observed under the MS treatment (*p* < 0.003). In addition, the MS treatment presented negative apparent digestibility for Ca and P. The apparent digestibility of K, like that of NDF, was not affected by treatments.

**Table 3 animals-15-03542-t003:** Production, nutritional composition, and nutritional efficiency in dual-purpose cows’ milk production from San Vicente del Caguán (Caquetá), supplemented with mineralized salt (MS), proteinized salt (PS), and concentrated feed (CO).

Response Variables		Treatments		SE	*p*
	CO	PS	MS		
Milk, L/cow/day	6.14	5.56	6.53	0.527	0.337
^1^ MF, %	5.35	5.44	4.84	0.352	0.468
MP, %	3.59	3.40	3.43	0.186	0.626
TMS, %	14.1	14.2	13.3	0.405	0.227
Lac, %	4.34	4.54	4.63	0.084	0.113
Ca	0.12	0.12	0.10	0.006	0.154
P	0.079	0.080	0.077	0.024	0.644
K	0.14	0.15	0.15	0.006	0.703
Milk deposition efficiency, %					
N	18.4	21.8	25.8	2.29	0.084
Ca	14.4 ^ab^	12.8 ^b^	18.1 ^a^	1.37	0.004
P	20.4 ^a^	13.6 ^b^	21.5 ^a^	1.72	0.004
K	5.94	5.81	6.48	0.83	0.723

Means in the same row, followed by different superscripts (^a^, and ^b^), are significantly different (*p* < 0.05); ^1^ MF: milk fat; MP: milk protein; TMS: total milk solids; Lac: lactose in milk; Ca; calcium; K: potassium; P: phosphorus; N: nitrogen; *p*: Statistical significance value; SEM: Standard error.

**Table 4 animals-15-03542-t004:** Correlations between the efficiency of Ca, P, K, and N deposition in milk with milk production and the intake of each nutrient.

Nutrient	Milk, L/Cow/Day
Ca	0.836
P	0.881
K	0.786
N	0.590

**Table 5 animals-15-03542-t005:** Concentration of exhaled CH4, volume of exhaled air, and enteric CH4 emissions in dual-purpose cows supplemented with the three evaluated treatments.

Response Variables	Treatments			SE	*p*
	CO	PS	MS		
^1^ CH_4_, ppm	1759	1732	1911	47.0	0.156
EXHA, L/cow/min	104	109	105	2.68	0.359
CH_4_, L/day	263	272	286	7.38	0.146
CH_4_, g/day	189	195	205	5.26	0.152
CH_4_, %GE	6.30 ^b^	8.23 ^a^	7.59 ^a^	0.48	0.030
CH_4_, L/L milk	45.9	52.8	50.2	6.13	0.719
CH_4_, g/L milk	32.8	37.8	35.9	4.37	0.717
CH_4_, g/FPCM	39.1	47.2	43.8	4.21	0.649
CH_4_, L/kg TDMI	40.4	37.3	32.9	2.87	0.133
CH_4_, g/kg TDMI	28.9	26.6	23.6	2.00	0.132

Means in the same row, followed by different superscripts (^a^, and ^b^), are significantly different (p < 0.05); ^1^ CH_4_, ppm: Parts per million of CH4 in exhaled air; EXHA, L/cow/min: Exhalation in liters per cow per animal; CH_4_, L/day: CH_4_ emissions in liters per day; CH_4_, g/day: CH_4_ emissions in grams per day; CH_4_, %GE: CH_4_ emission as percent of gross energy intake (Y_m_); CH_4_, L/L milk: CH_4_ intensity in liters per liter of milk; CH_4_, g/L milk: CH_4_ intensity in grams per liter of milk; CH_4_, g/FPCM: CH_4_ emissions in g/L of milk corrected for fat and protein; CH_4_, L/kg TDMI: CH_4_ emissions in liters per kilogram of total dry matter intake; CH_4_, g/kg TDMI: CH_4_ emissions in grams per kg of total dry matter intake; SE: Standard error.

## Data Availability

Dataset available on request from the authors.

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
