# Peer review of "Effect of Dietary Supplementation on Milk Nutrient Deposition and Enteric Methane Emission in Dual-Purpose Cows from the Colombian Amazon"

_animals, 2025, doi:10.3390/ani15243542_

Round 1
Reviewer 1 Report
Comments and Suggestions for Authors
The manuscript is of great importance; however, there is information that should be included to improve it.

Author Response
Title
Comments 1: The quality of the milk was not taken into account
Response 1: it was corrected
Introduction
Comments 2: lines 77 -84: What is quoted in this paragraph is very similar to what is quoted in the following paragraph and, to make it sound repetitive, consider restructuring this information.
Response 2: it was corrected
Comments 3: lines 97 - 112: I believe this information is not relevant to include in the manuscript, as it does not justify why the research was conducted (delete)
Response 3: it was corrected
Comments 4: line 141: If only 18 days of the experiment were evaluated, and it is stated that the evaluation took place between March and May, I consider it pertinent to include the exact evaluation date; for example: from March 13 to May 14.
Response 4: These dates include the preparation of supplemental foods, external marker, selection of experimental animals, the displacement of equipment and researchers to farm and, the experimental periods with animals. The initiation of first experimental period was April 19 and the date of ending of last period was May 6.
Comments 5: lines 157 – 158: If rotational grazing was used, what was the area used in each paddock?
Response 5: As is the case in most cattle herds in tropical conditions, the area of ​​the paddocks offered to the cows daily was variable due to the variability in the availability of forage that occurred as a consequence of differences in soil quality and the management that had previously been established on each paddock by the herd owner. The average of area used daily was approximately of 2200 m2.
Comments 6: line 165: If it is said that the supplementation was based on the nutritional composition of the pastures, how can we know that the cows consumed more of one grass than another, in the sense that Mombasa grass is erect and reaches heights of up to 100 cm and star grass averages heights of 50 cm? I believe that a botanical composition should have been carried out to determine the percentage of each grass species present in the pasture.
Response 6: the nutritional composition of pastures previously of the preparation of supplements was summited by SOMEX S.A.S. During the final days of the experimental periods, samples were taken from each pasture and their presence in the paddocks was estimated. Mombasa grass and star grass were found in high proportions, with other grasses such as Humidicola and Paspalum notatum in smaller proportions, and Cyperus rotundus was the most common weed but with lower presence (Here 3 photos to illustrate the conditions of paddocks).
I also consider it important to include information on the pasture height at the time the animals entered and the remaining height upon their removal, and based on this, determine the quality of the forage consumed.
Response 6: this information does not was taken.
On the other hand, it is necessary to include how the sampling was carried out to determine the quality of the pastures.
Response 6: it was corrected.
Comments 7: Table 1: It must be: 1MP, as indicated at the bottom of the table
Response 7: it was corrected.
Comments 8: line 219: It was previously stated that they were Holstein × Gyr
Response 8: it was corrected.
Comments 9: line 265: place in superscript
Response 8: it was corrected in all paper.
Comments 9: Table 2: Place the correct form in which the treatment will be represented (CO, CO1 o 1CO)
Response 9: it was corrected
Comments 10: lines 582 - 588: The conclusion should address the objective; review and adapt it.
Response 10: it was corrected

Reviewer 2 Report
Comments and Suggestions for Authors
I invite the authors to respond to these questions:
Introduction
Lines 51–72: The description of cattle production systems in Colombia is well presented and supported by appropriate data. However, this subsection is overly detailed for an introduction. Consider reducing the numerical percentages unless they are directly relevant to the study rationale.
Lines 62–64: The statement regarding the role of the dual-purpose system (DPS) in national milk production is important and should be emphasized earlier to highlight the relevance of studying DPS.
I suggest that the authors group lines 51–74 into a shorter and more focused subsection, highlighting why the DPS is important for Colombia and why nutritional interventions matter.
Lines 82–90: The discussion on C3 vs. C4 metabolic pathways is scientifically accurate, but references should be updated if more recent evidence exists. These lines effectively justify the need for supplementation strategies; however, some sentences are repetitive and can be tightened for clarity.
Lines 97–104: The historical background on the greenhouse effect (Fourier, Murphy, Phillips) is overly detailed for an introduction. This level of historical narrative is unnecessary and disrupts the flow.
Lines 128–132: The objective is clearly stated; however, consider stating the objectives earlier (after identifying the main research gap) to improve coherence.
Materials & Methods
Lines 173–179, 280–289: The experimental description indicates 24 cows entered the experiment in four 12-day experimental periods with six cows/period (Lines 173–176). It is not clear whether this is a cross-over, replicated Latin square, or a repeated measures/blocking design. The sentence "cows were randomly assigned to one of three treatments using a completely randomized block design (Figure 1)" (Lines 158–160) conflicts with the later description of periods/blocks.
Lines 280–289: The current linear model is simplistic and does not reflect the blocking/period structure described earlier, nor random effects or repeated measures. I invite the authors to replace with a full mixed-model description and indicate whether cow is treated as random effect, how repeated measures were handled (milk measured multiple days), which covariance structure was used and how it was selecte, Alpha-level for significance and multiple comparisons adjustment, and software package and version (SAS 9.x? R 4.x? which procedure?)
Lines 151–156: There is no statement justifying sample size (n = 24). Provide rationale or a post-hoc power calculation to show the study is adequately powered to detect the minimum biologically relevant difference for milk and CHâ‚„ outcomes.
Discussion
Lines 339–579: The Discussion is disproportionately long relative to the study and contains extensive literature summarization. For instance, the authors list multiple studies reporting “no effect” of supplementation on DMI or milk composition (Lines 367–446), many of which contribute little additional insight. I invite the authors to condense repeated patterns of results into synthesized statements rather than listing each study individually.
Line 352–357: ADMD increased in the CO treatment, but the authors only mention possible variability in the relationship between ADMD and TDMI. They do not explain why starch-based supplementation may improve digestible fraction yet fail to translate into greater intake.
Lines 375–386: The discussion of NDFD lacks mechanistic links (e.g., rumen pH, substitution rate, microbial dynamics, passage rate). The interpretation needs better depth.
Lines 388–407: Negative digestibility of Ca and P is reported, but the section is overwhelmingly descriptive and includes too many tangential citations. A focused, mechanistic explanation is needed (e.g., endogenous secretions, mineral-binding, passage rate).
Line 463: I invite the authors to refer to this paper for their statement:
Effects of sulla flexuosa hay as alternative feed resource on goat’s milk production and quality. Animals, 13(4), 709. https://doi.org/10.3390/ani13040709
Microalgae supplementation improves goat milk composition and fatty acid profile: A meta-analysis and meta-regression. Archives Animal Breeding, 68(1), 223-238. https://doi.org/10.5194/aab-68-223-2025
The methane section (Lines 511–579) is detailed, but the authors do not connect methane results with earlier findings on digestibility, fiber intake, enzyme activity, or forage quality. Also, regression equations (Lines 559–562) are cited without evaluating whether they fit the study’s data. I invite the authors to integrate CHâ‚„ results with the nutritional discussion and evaluate how the data compare to predictive models.
Author Response
Introduction
Comments 1: Lines 51–72: The description of cattle production systems in Colombia is well presented and supported by appropriate data. However, this subsection is overly detailed for an introduction. Consider reducing the numerical percentages unless they are directly relevant to the study rationale.
Response 1: it was corrected
Comments 2: Lines 62–64: The statement regarding the role of the dual-purpose system (DPS) in national milk production is important and should be emphasized earlier to highlight the relevance of studying DPS.
I suggest that the authors group lines 51–74 into a shorter and more focused subsection, highlighting why the DPS is important for Colombia and why nutritional interventions matter.
Response 2: it was corrected
Comments 3: Lines 82–90: The discussion on C3 vs. C4 metabolic pathways is scientifically accurate, but references should be updated if more recent evidence exists. These lines effectively justify the need for supplementation strategies; however, some sentences are repetitive and can be tightened for clarity.
Response 3: it was corrected
Comments 4: Lines 97–104: The historical background on the greenhouse effect (Fourier, Murphy, Phillips) is overly detailed for an introduction. This level of historical narrative is unnecessary and disrupts the flow.
Response 4: it was corrected
Comments 5: Lines 128–132: The objective is clearly stated; however, consider stating the objectives earlier (after identifying the main research gap) to improve coherence.
Response 5: we consider that the target is well placed in the current position of the introduction. Materials & Methods
Comments 6: Lines 173–179, 280–289: The experimental description indicates 24 cows entered the experiment in four 12-day experimental periods with six cows/period (Lines 173–176). It is not clear whether this is a cross-over, replicated Latin square, or a repeated measures/blocking design.
Response 6: We appreciate the reviewer comments and suggestions. To clarify, the design corresponds to a randomized complete block design (RCBD), where the four 12-day experimental periods functioned as temporal blocks, and each cow received only one treatment throughout the study. This information was rewritten and added in the experimental design section.
Comments 7: Lines 280–289: The current linear model is simplistic and does not reflect the blocking/period structure described earlier, nor random effects or repeated measures. I invite the authors to replace with a full mixed-model description and indicate whether cow is treated as random effect, how repeated measures were handled (milk measured multiple days), which covariance structure was used and how it was selecte, Alpha-level for significance and multiple comparisons adjustment, and software package and version (SAS 9.x? R 4.x? which procedure?)
Response 7: The linear model used here included the blocking structure but not the repeated measures effect because data collection did not include this sampling structure. Now, because the cows do not present differences in live weight, days in milk and initial milk production was used as a covariate, we considered that it was not necessary to include the cows as a random effect in the analysis.
This was the SAS syntax used:
proc glm data=mydata;
class treatment block;
model response = treatment block covariate;
lsmeans treatment / pdiff adjust=tukey;
run;
quit;
Comments 8: Lines 151–156: There is no statement justifying sample size (n = 24). Provide rationale or a post-hoc power calculation to show the study is adequately powered to detect the minimum biologically relevant difference for milk and CHâ‚„ outcomes.
Response 7: We thank the reviewer for your suggestion. We added a full post hoc statistical power evaluation using the observed variability in the primary response variable (daily methane production). You can track the highlighted change in the sample size section in the Materials and Methods section.
Discussion
Comments 9: Lines 339–579: The Discussion is disproportionately long relative to the study and contains extensive literature summarization.
For instance, the authors list multiple studies reporting “no effect” of supplementation on DMI or milk composition (Lines 367–446), many of which contribute little additional insight. I invite the authors to condense repeated patterns of results into synthesized statements rather than listing each study individually.
Response 9: it was corrected
Comments 10: Line 352–357: ADMD increased in the CO treatment, but the authors only mention possible variability in the relationship between ADMD and TDMI. They do not explain why starch-based supplementation may improve digestible fraction yet fail to translate into greater intake.
Response 10: it was corrected
Comments 11: Lines 375–386: The discussion of NDFD lacks mechanistic links (e.g., rumen pH, substitution rate, microbial dynamics, passage rate). The interpretation needs better depth.
Response 11: it was corrected
Comments 12: Lines 388–407: Negative digestibility of Ca and P is reported, but the section is overwhelmingly descriptive and includes too many tangential citations. A focused, mechanistic explanation is needed (e.g., endogenous secretions, mineral-binding, passage rate).
Response 12: it was corrected
Comments 13: Line 463: I invite the authors to refer to this paper for their statement: Effects of sulla flexuosa hay as alternative feed resource on goat’s milk production and quality. Animals, 13(4), 709. https://doi.org/10.3390/ani13040709
Microalgae supplementation improves goat milk composition and fatty acid profile: A meta-analysis and meta-regression. Archives Animal Breeding, 68(1), 223-238. https://doi.org/10.5194/aab-68-223-2025
Response 13: we do not understand how utilize the information of these papers on the text.
Comments 14: The methane section (Lines 511–579) is detailed, but the authors do not connect methane results with earlier findings on digestibility, fiber intake, enzyme activity, or forage quality.
Response 14: it was corrected
Comments 15: Also, regression equations (Lines 559–562) are cited without evaluating whether they fit the study’s data. I invite the authors to integrate CHâ‚„ results with the nutritional discussion and evaluate how the data compare to predictive models.
Response 15: We affirm that “Some authors have reported regression equations in which milk production is associ-ated with CH4 production. Thus, Van Amburgh et al., [115] reported that CH4 produc-tion can be predicted from milk production by equation CH4 (kg/d) = 0.004 × Milk (kg/d) + 0.43. For its part, Niu et al., [116] reported the equation CH4 (g/d) = 2.27 x Milk (kg/d) + 314”. However, these equations were derived from a database with 159 treatments in Van Amburgh et al. [115] and with 1212 observations in the publication by Niu et al. [116], in which the data from this experiment fall outside the range of Van Amburgh et al. [115] but are at the lower end of the data from Niu et al. [116]. Therefore, we consider it inappropriate to compare the data from the present experiment with the values ​​estimated by the above equations. For this reason, we deem it necessary to remove the statement about the regression equations from the text.

Round 2
Reviewer 2 Report
Comments and Suggestions for Authors
Response 6 – Experimental Design Clarification
Thank you for clarifying that the study used an RCBD.
However, because cows remain the experimental units in a blocked design:
Please specify the blocking factor explicitly in the experimental design description.
Since each cow received only one treatment, please clarify whether cows were randomly allocated to treatments within each block, and whether the block size was balanced.
Response 7 – Statistical Model
Several issues remain unresolved:
Repeated Measures: You state that repeated measures were not included because “data collection did not include this sampling structure,” yet milk production was measured on multiple days.
Please clarify how many milk measurements per cow were used and how they were aggregated.
If daily measures were averaged per period, please state this explicitly.
Random Effects: Your justification for not including cow as a random effect (“no differences in LW, DIM, or initial milk”) is not statistically correct.
Random effects are used to account for individual variability, not for observed differences at baseline. Please reconsider whether a mixed model with cow as random effect is statistically more appropriate or justify in more detail why GLM is adequate for this design.
Response 13 – Cited Papers
You mention you “do not understand how to utilize the information” from the suggested papers.
These papers should be added as references for this paragraph “These results may be due to the fact that the response to supplementation in the production and composition of milk is variable and depends on multiple factors such as the genetics of the animal, the stage of lactation, the availability and quality of the pastures, as well as management practices “. They are about the factors that affect the variation of milk composition.
Response 15 – Removal of Regression Models
Your argument for removing the predictive equations is understandable, but it raises two questions:
Even if the equations do not fit your data range, is there value in briefly stating how your methane values compare qualitatively (higher/lower/similar) to typical values in dairy cows?
This would help contextualize your results.
If you remove the regression equations entirely, please ensure the methane section still integrates with the nutritional discussion, as requested earlier.
Author Response
We appreciate your comments and suggestions, as they have strengthened the document and made it more understandable for readers.
Below we respond to your new comments.
Comments to Response 6 – Experimental Design Clarification
Thank you for clarifying that the study used an RCBD.
However, because cows remain the experimental units in a blocked design:
Please specify the blocking factor explicitly in the experimental design description.
Since each cow received only one treatment, please clarify whether cows were randomly allocated to treatments within each block, and whether the block size was balanced.
Response 6: In was corrected in the manuscript
Comments to Response 7 – Statistical Model
Several issues remain unresolved:
Repeated Measures: You state that repeated measures were not included because “data collection did not include this sampling structure,” yet milk production was measured on multiple days.
Please clarify how many milk measurements per cow were used and how they were aggregated.
If daily measures were averaged per period, please state this explicitly.
Random Effects: Your justification for not including cow as a random effect (“no differences in LW, DIM, or initial milk”) is not statistically correct.
Random effects are used to account for individual variability, not for observed differences at baseline. Please reconsider whether a mixed model with cow as random effect is statistically more appropriate or justify in more detail why GLM is adequate for this design.
Response 7 It was corrected in the manuscript. The SAS syntax used was:
PROC mixed data =a;
CLASS TTO BQUE COW;
MODEL LW = TTO BQUE/ddfm=satterth;
RANDOM COW/subject=COW;
LSMEANS TTO/PDIFF=all; RUN;
We have attached the results of the SAS analysis.
Comments to Response 13 – Cited Papers
You mention you “do not understand how to utilize the information” from the suggested papers.
These papers should be added as references for this paragraph “These results may be due to the fact that the response to supplementation in the production and composition of milk is variable and depends on multiple factors such as the genetics of the animal, the stage of lactation, the availability and quality of the pastures, as well as management practices “. They are about the factors that affect the variation of milk composition.
Response 13: It was corrected in the manuscript
Comments to Response 15 – Removal of Regression Models
Your argument for removing the predictive equations is understandable, but it raises two questions:
Even if the equations do not fit your data range, is there value in briefly stating how your methane values compare qualitatively (higher/lower/similar) to typical values in dairy cows?
This would help contextualize your results.
If you remove the regression equations entirely, please ensure the methane section still integrates with the nutritional discussion, as requested earlier.
Response 15: First, although the predictive equations of Van Amburgh et al. [115] and Niu et al. [116] are not appropriate for estimating CHâ‚„ in our experiment, the values ​​found in this experiment fall within the typical ranges reported for cows in tropical systems. Therefore, we include in the manuscript a brief comparison indicating that the emissions observed in this study (196 ± 18 g/cow/d) are within the lower to middle range described for dual-purpose grazing cows (180–300 g/cow/d), as reported by Primavesi et al. (2004), Aguilar-Pérez et al. (2011), and Rivera et al. (2015), among others. This contextualization allows the reader to understand our results without extrapolating equations developed under production conditions, diets, and intake levels substantially different from those of our study system. Secondly, the manuscript includes a discussion of the relationship between the nutritional composition of the pasture and feed supplements and methane emissions.

Round 3
Reviewer 2 Report
Comments and Suggestions for Authors
The authors responded to all my comments.